# Furbellow (Brown Algae) Extract Increases Lifespan in *Drosophila* by Interfering with TOR-Signaling

**DOI:** 10.3390/nu12041172

**Published:** 2020-04-22

**Authors:** Yang Li, Renja Romey-Glüsing, Navid Tahan Zadeh, Jakob von Frieling, Julia Hoffmann, Patricia Huebbe, Iris Bruchhaus, Gerald Rimbach, Christine Fink, Thomas Roeder

**Affiliations:** 1Department of Molecular Physiology, Kiel University, D-24098 Kiel, Germany; baskerville_cap@hotmail.com (Y.L.); renja.romey@uni-hamburg.de (R.R.-G.); ntahanzadeh@zoologie.uni-kiel.de (N.T.Z.); jfrieling@zoologie.uni-kiel.de (J.v.F.); julia.hoffmann2@gmx.de (J.H.); cfink@zoologie.uni-kiel.de (C.F.); 2College of Life Sciences, Qingdao University, Qingdao 266071, China; 3Department of Food Sciences, Kiel University, 24098 Kiel, Germany; huebbe@foodsci.uni-kiel.de (P.H.); rimbach@foodsci.uni-kiel.de (G.R.); 4Bernhard-Nocht-Institute for Tropical Medicine, D-20359 Hamburg, Germany; bruchhaus@bnitm.de; 5DZL, German Center for Lung Research, ARCN, D-24098 Kiel, Germany

**Keywords:** *Saccorhiza polyschides*, brown algae, TOR signaling, *Drosophila melanogaster*, *Imp-L2*, insulin signaling, lifespan

## Abstract

Algal products are well known for their health promoting effects. Nonetheless, an in depth understanding of the underlying molecular mechanisms is still only fragmentary. Here, we show that aqueous furbelow extracts (brown algae, *Saccorhiza polyschides*) lengthen the life of both sexes of the fruit fly *Drosophila melanogaster* substantially, if used as nutritional additives to conventional food. This life prolonging effect became even more pronounced in the presence of stressors, such as high-fat dieting of living under drought conditions. Application of the extracts did not change food intake, excretion, or other major physiological parameters. Nevertheless, effects on the intestinal microbiota were observed, leading to an increased species richness, which is usually associated with healthy conditions. Lifespan extension was not observed in target of rapamycin (TOR)-deficient animals, implying that functional TOR signaling is necessary to unfold the positive effects of brown algae extract (BAE) on this important trait. The lack of life lengthening in animals with deregulated TOR signaling exclusively targeted to body fat showed that this major energy storage organ is instrumental for transmitting these effects. In addition, expression of *Imaginal morphogenesis protein-Late 2* (*Imp-L2*), an effective inhibitor of insulin signaling implies that BAE exerts their positive effects through interaction with the tightly interwoven TOR- and insulin-signaling systems, although insulin levels were not directly affected by this intervention.

## 1. Introduction

Lengthening life and slowing the process of aging is a dream of mankind. Only very few interventions are known to increase lifespan reproducibly. Among them, dietary or caloric restriction is outstanding, because its efficacy has been shown in a great number of different animal species of increasing biological complexity ranging from yeast to mammals [1,2,3]. Despite its great potential, its acceptance in the population is low, presumably caused by the strict and lifelong diet that is required to unfold its positive effects. Thus, alternative strategies to lengthen life are highly desirable. This applies especially to those interventions that mimic dietary restriction without the need of actual dieting [4]. Prototypical for such an intervention is the consumption of rapamycin, which is a macrolide with potent immunosuppressive activities, first isolated from *Streptomyces hygroscopicus* [5]. Rapamycin has been shown to increase lifespan in different models comprising yeast, worms, flies, and mice, while an anticipated positive effect on human lifespan is still heavily debated [6,7,8]. Mechanistically, this type of dietary restriction mimetic interacts with signaling systems such as the TOR- and the insulin-signaling pathways, which are highly relevant for transducing the effects of dietary restriction (DR) into lifespan prolongation [6,9,10].

Recent efforts identified some additional compounds that have the potential to lengthen life. Alpha-ketoglutarate is one example of a compound that also interferes with TOR activity to increase lifespan [11]. In order to identify these compounds, suitable screening systems are mandatory. Only two models attained substantial recognition, the soil nematode *Caenorhabditis elegans* and the fruit fly *Drosophila melanogaster* [7,12]. Although the first use of the fly in studies focusing on the effects of interventions on lifespan dates almost 70 years back [13], the majority of studies employed *C. elegans*, as it is easier to use in high-throughput formats [14]. Nevertheless, the fruit fly offers several advantages compared with *C. elegans* comprising an organ composition and nutritional physiology that shares considerable similarities with those of humans [15].

Screening compound libraries either using *C. elegans* or *Drosophila* as read-out tools offers several advantages, while some drawbacks have to be kept in mind. Substances that unfold their lifespan-prolonging effects only in combination with other compounds are usually not found and the development from a biologically active compound to a marketable drug is very long and expensive. Plant extracts as the first screening tool offer several advantages as they have combinations of compounds that have been tailored for specific functional goals during evolution and they offer an almost inexhaustible source for biologically active compounds. Algae are long known as important components of diets in coastal populations around the world [16]. Especially in east Asian countries such as Japan, Korea, or East China, algae are important parts of the diet. One impressive example highlighting the relevance of algae as a diet component is the famous Okinawan diet, which has a high abundance of functional marine foods. In particular, these functional components are thought to foster healthy aging [17].

The major goal of this current study is to find a novel plant or algal extract with a lifespan-prolonging effect. For this, we tested a large number of different extracts using the fruit fly *Drosophila melanogaster* for their ability to increase lifespan. We identified life lengthening effects of an aqueous brown algae extract (*Saccorhiza polyschides*) and showed that this effect is mediated through interaction with the TOR signaling pathway.

## 2. Materials and Methods

### 2.1. Fly Husbandry and Strains

Flies were maintained as already described before [18,19]. The following fly strains were used for the study: w^1118^ (Bloomington stock ID #5905), w^1118^; P{Switch1}106-GS (gift from R. Kühnlein, Göttingen), UAS-Tor.TED (Bloomington stock ID #7013), UAS-Tor.WT (Bloomington stock ID #7012), w^1118^; UAS-dcr2; dilp2-gd2HF(attp2), dilp2-15-1-HA-Gal4 (gift from S. Kim, Stanford). All lifespan studies used mated flies from the genotype *w^1118^*. We chose this strain since it is the background strain of most transgenes and has been successfully used in a great variety of lifespan analyses. Animals were separated by sex 48 h after hatching. Afterwards, flies of the experimental groups were brought into contact with the brown algae extract (BAE) for the rest of their adult life. Vials were changed twice a week. Dead flies were counted four times a week. Flies were kept under constant conditions (25 °C, 65% humidity, and a day/night rhythm of 12 h/12 h). Control medium contained 5% (*w*/*v*) yeast extract (Becton Dickinson, Heidelberg, Germany), 5% (*w*/*v*) sucrose, 8.6% cornmeal, 0.5% (*w*/*v*) agar, 0.03% (*v*/*v*) propionic acid, and 0.3% (*v*/*v*) methyl-4-hydroxybenzoat. Extracts were added to the medium at reduced temperatures directly before pouring the vials. For experiments with high-fat diets, the normal medium was supplemented with 10% of palm fat.

Experiments employing the GeneSwitch system were performed as described recently [19]. Freeze-dried aqueous extracts (*Saccorhiza polyschides*) were generously made available by Coastal Research Management (CRM; Kiel, Germany). The major secondary metabolites of brown algae extract (BAE) are dieckol, eckol, bieckol, other polyphenols, ergosterol, and fucoxanthin. A detailed analysis of the major components of this extract as evaluated by liquid chromatography coupled to mass spectrometry (LC-MS) was previously performed [20].

### 2.2. Statistical Analysis

All statistics were performed using GraphPad Prism version 6.00 for Windows (GraphPad Software, La Jolla, CA, USA). The data was tested for distribution by using the Shapiro–Wilk normality test. Parametric data were analyzed with unpaired Student’s *t*-test and non-parametric data with Mann–Whitney test. Data in Figures 3, 4, and 6 were analyzed with Dunnett’s test. Lifespan and starvation survival data were analyzed by log rank (Mantel–Cox) test.

### 2.3. Body Fat Quantification

Total body triacylglycerols (TAGs) in flies were measured using the coupled colorimetric assay (CCA) method as described previously [19,21]. Glyceryl trioleate (T7140, Sigma-Aldrich, Taufkirchen, Germany) served as the TAG standard. Five females (or 8 males) per group were weighted and homogenized in 1 mL 0.05% Tween-20 using a Bead Ruptor 24 (BioLab Products, Bebensee, Germany). Homogenates were heat-inactivated at 70 °C for 5 min, centrifuged, and incubated with triglyceride solution (Fisher Scientific, Schwerte, Germany) at 37 °C for 30 min with shaking. The absorbance was read at 562 nm and the quantity estimated using the standard curve. Each measurement was performed with at least three biological replicates.

### 2.4. Metabolic Rate Determination

Relative quantification of metabolic rates was performed according to Yatsenko and colleagues [22]. In brief, the metabolic rate of 3 adult flies per vial was measured for 2 h using respirometry. Data were calculated based on the volume of CO_2_ production during the test, which are presented as μL per hour per fly. Data of 5 independent biological replicates were combined.

### 2.5. Hemolymph Glucose Measurement

The glucose measurement was performed using the Glucose (HK) Assay Kit (GAHK-20, Sigma-Aldrich, Taufkirchen, Germany) with minor modifications. Hemolymph samples were pooled from 15–20 flies of each genotype as described [23]. The sample was diluted 1:10 in deionized water and added to 50 μL glucose standard solution and incubated for 15 min at room temperature (RT). Total glucose was recorded by measuring absorbance at 340 nm.

### 2.6. Drosophila Insulin-Like Protein 2 (dILP2) Measurements

To quantify circulating as well as total dILP2 levels, the method described in Park et al. was used with some modifications [24]. Flies carrying the FLAG- and HA-tagged version of dILP2 allowed the quantification of circulating and total amounts of dILPs. For hemolymph extraction, the abdomen of 30 flies per group were pricked with a fine steel needle. Afterwards, the flies were transferred into 0.5 mL reaction with holes on ice. The 0.5 mL reaction tube was placed within a 1.5 mL reaction tube and centrifuged at 3000× *g* at 4 °C for 1 min. The yielded hemolymph was used for ELISA. For total dILP2 quantification, 5 flies were transferred into a 1.5 mL reaction tube with 250 µL PBT (phosphate-buffered saline (PBS) with 1% Triton-X100) and ground with a pestle. The tube was vortexed for 5 min and centrifuged for 5 min at 10,000× *g*. Afterwards, 50 µL of the supernatant was used for ELISA. For ELISA, an anti-FLAG^®^ High Sensitivity M2 coated 96-well plate was used. The samples were further processed exactly as described in Park et al. [24].

### 2.7. Microbiota

Analysis of the composition of the intestinal microbiota was performed using feces samples of small groups of adult fruit flies essentially as described earlier [25,26]. In brief, feces and intestines of 20 individuals per replicate were collected with a sterile swab, respectively. DNA was extracted with the DNeasy Blood and Tissue Kit (Qiagen, #69504) following the manufacturer’s protocol for “Pretreatment for Gram-Positive Bacteria”, followed by the protocol for “Purification of Total DNA from Animal Tissues”. DNA was eluded in 100 µL sterile elution buffer and stored at −20 °C until sequencing. The bacterial variable regions 1 and 2 of 16S rRNA genes were amplified as described by Rausch et al. [27,28]. The amplicons were sequenced using the Illumina MiSeq with 2 × 300 bp paired-end sequencing. Assembling, quality filtering, and chimera detection were performed by using Qiagen CLC Genomics Workbench 20 Microbial Genomic Module. Filtered sequences were aligned 97% to the Greengenes reference database (gg_13_8 release) with a similarity score of 80%.

## 3. Results

In order to identify plant or algae extracts that have a positive effect on lifespan, we used the local PECKISH (Plant Extract Collection Kiel in Schleswig Holstein) extract library that was described in detail earlier [29]. For this, we used the fruit fly *Drosophila melanogaster* as a screening tool and quantified the effects of supplementation of normal diets with these extracts on lifespan. Cohorts of mated male and female flies of the *w^1118^* strain were used for this purpose. We designed the experiments with more than 100 animals per biological replicate each and with multiple independent replication rounds. Those extracts that showed lifespan enhancing effects in the first screening round were transferred to the second round with independent experiments, and only those positive ones were then tested in a third round, which was also used for quantification of the data. Almost all extracts had no or even a negative effect on lifespan. In contrast, one alga extract increased lifespan substantially and reproducibly (Figure 1). This extract was generated from the brown algae *Saccorhiza polyschides*. This positive effect on lifespan was seen in both sexes (Figure 1A,B). Two different algae extract concentrations, 0.1% and 0.5%, were tested. For 0.1% extract concentration, the increases in median lifespans were from 65 d to 70 d in females (*p* = 0.0006) and from 57 d to 63 d in males (*p* < 0.0001); and for 0.5% extract concentration the median lifespans were 67 d for females (*p* = 0.023) and 64 d for males (*p* = 0.0044). The maximal lifespans (10% oldest animals) were also increased in females from 84.2 ± 0.81 d in controls to 92 ± 1.12 d in 0.1% BAE-treated flies and 94 ± 0.85 d in 0.5% BAE-treated flies (*p* < 0.0001 for both comparisons). In males, maximal lifespans increased from 75.91 ± 0.89 (control) to 81.27 ± 0.36 (0.1% AE, *p* < 0.0001) and 76.91 ± 0.44 (0.5% AE, not significant (n.s.)), respectively.

Under stressful conditions, the positive effects on lifespan became more pronounced. Feeding a high-fat diet together with 0.1% BAE increased the median lifespan from 26 d (only high-fat diet) to 31 d in females (*p* < 0.001; Figure 1C) and from 25 d to 30 d in males (*p* < 0.001; Figure 1D). To evaluate if this effect of this brown algae extracts is also long-lasting, control animals were placed the entire experimental time on a high-fat diet, whereas the treated animals were placed for the first 10 days of the experiment on a high-fat diet in combination with 0.1% BAE, but removed and placed back to a high-fat diet without algae extract for the rest of the experiment. Even under these conditions, lifespans were very similar in BAE-treated animals if compared with those animals that were treated their entire adult life with BAE. Female controls had a lifespan of 27 d (Figure 1E) and treated ones a lifespan of 37 d (*p* < 0.001); the increase seen in males was from 28 d (control) to 39 d (0.1% BAE, *p* < 0.001, Figure 1F).

Other stressful conditions were also better tolerated by animals that experienced food enriched with BAE. Under drought stress (20% humidity, Figure 2), the lifespans of females increased from 52 d (control) to 61 d (0.1% BAE, *p* < 0.001) and 59 d (0.5% BAE, *p* = 0.0066), respectively (Figure 2A). In males, the median lifespans were 59 d (control), 61 d (0.1% BAE, *p* < 0.001), and 64 d (0.5% BAE, *p* < 0.001), respectively (Figure 2B). Flies treated with brown algae extracts were also more starvation resistant than matching controls. The median survival time of females was 42 h, while those females treated with 0.1% AE survived for 49 h and those treated with 0.5% BAE for 51 h (Figure 2C, *p* < 0.001 for both comparisons). In males, the survival time increased from 42 h (control) to 44 h (0.1% BAE, *p* < 0.01; 0.5% BAE, *p* < 0.05, Figure 2D).

To exclude confounding effects of AE addition to the diet on feeding rates, we quantified the daily intake of nutrients by using capillary feeder (CAFE) experiments [30]. Both, treatment with 0.5% BAE and with 0.1% BAE did not alter food intake at all (Figure 3A), which thus excludes confounding effects caused by reduced food intake leading to a caloric restriction phenotype. Moreover, treatment with these brown algae extracts had no effect on lifetime fecundity (Figure 3B) or overall physical activity (Figure 3C). Furthermore, the metabolic rates of the corresponding flies were neither altered in females (Figure 3D) nor in males (Figure 3E), even if the analysis was performed after different times of exposure to the BAE.

Whereas the body weights of males were slightly increased in 0.5% BAE-treated animals (Figure 4A), the corresponding triacylglycerol levels were reduced (Figure 4B). For females, neither effects on body weight nor on triacylglycerol levels were observed (Figure 4A,B). Regarding the effects on hemolymph glucose levels, we observed a sex-specific effect, where females showed a slight but statistically significant increase, whereas males showed a slight decrease in glucose levels (Figure 4C). To learn more about the underlying mechanisms, we measured the levels of circulating insulin (dILP2) levels in response to confrontation with the algae extract using transgenic flies that have been characterized recently [18,24]. After one week of confrontation with the extracts, differences between treated and untreated animals were not observed either for the total dILP2 (Figure 4D) or for the circulating dILP2 levels (Figure 4E).

To elucidate the mode of action of the life lengthening effects of BAE, we performed studies with animals defective in the TOR-signaling pathway as the most prominent pathway relevant for lifespan extending interventions (Figure 5). The median lifespan of TOR-deficient flies (86.5 d) was not significantly changed after supplementation with 0.1% BAE (84.5 d) in females (Figure 5A, *p* = 0.664), as well as in males, where controls showed a survival time of 85 d and BAE-treated ones showed a survival time of 87 d (Figure 5B, *p* = 0.285). Animals that experienced ectopic overexpression of a dominant negative TOR allele in the fat body only (UAS-Tor.TED), which was achieved using the GeneSwitch system that was activated by mifepristone [31], showed no positive effect on lifespan. Lifespans dropped from 99 d (controls) to 94 d (0.1% BAE) in females (*p* < 0.01, Figure 5C) and from 82 d (control) to 76 d (0.1% BAE) in males (*p* < 0.01, Figure 5D).

Furthermore, we analyzed changes in the transcript levels of selected target genes implicated to be relevant for various aspects of lifespan extensions. We tested the transcript levels of these candidate genes after 7 d (Figure 6A,B) and 14 d (Figure 6C,D) of treatment with 0.1% and 0.5% AE in both sexes. For a variety of the genes under investigation, significant changes were observed. In females (e.g., the glut1, puckered, ILP8, and the Imp-L2) genes were upregulated after 7 d of AE exposure, whereas after 14 d the regulation was strongly reduced. On the other hand, in males, we observed small, but significant changes after 7 d, but Imp-L2 levels were enhanced most strongly after 14 d (Figure 6B,D).

To evaluate if the effects of the extracts might also be caused by an effect on the microbiota, we evaluated their composition in response to prolonged treatment with BAEs. We analyzed both the microbiota derived from the intestine (Figure 6E–G) and that from the feces (Figure 6H–K). The 16S-based analysis revealed small differences in the composition of the microbiota as shown by the principal coordinates analysis (PCoA) plots (Figure 6E,H) and the bar chart graphs (Figure 6F,I). Significant differences were observed in the species richness of the fecal samples (the samples of the intestines show in the same direction, without being significantly different) as demonstrated by the alpha diversities (Figure 6G,K).

## 4. Discussion

Nutritional components have a great impact on aging [32]. This very simple statement applies both to specific food components and to the total calorie intake. On the one hand, these are life-span prolonging interventions such as caloric or dietary restriction [33]. On the other hand, fat- and carbohydrate-rich Western diets lead to a reduced life expectancy [34]. In the current study, we showed that addition of brown algae extracts to normal food used for feeding in *Drosophila* was sufficient to increase lifespan. Although the effects were seemingly small, a 10% increase in lifespan is not only statistically significant, but also biologically relevant. This increase was seen in both sexes and became even more pronounced under stressful conditions such as high-fat dieting. Dietary use of seaweed has been associated with a variety of health associated parameters, comprising reduced obesity rates, reduced coronary heart diseases, enhanced neuroprotection and even reduced cancer entities [35]. Moreover, extracts from algae have been shown to normalize a variety of health-associated parameters in high-fat dieting rodents [36]. Nevertheless, direct lifespan prolongation induced by algae extracts has only been reported occasionally [37].

The positive effects of brown algae extracts are not restricted to *Drosophila*. Very recently, it was shown that the same extract counteracts diet-induced obesity in mice [20]. The effects were multifactorial, but the effects on the intestinal microbiota appeared to be highly relevant. The composition of the microbiota and their ability to produce certain metabolites has recently been shown to have a major impact on lifespan [38]. Here, we also show alterations in the composition of the microbiota, especially the increased alpha-diversity, which is indicative of increased bacterial species richness associated with health-promoting effects [39]. Moreover, administration of brown algae extracts was shown to influence glucose and triglyceride levels in rabbits [40], which is also completely in-line with our results.

Apparently, this alga extract-induced lengthening of life was due to an interaction with the TOR signaling pathway, which takes a central position in aging associated processes [9]. TOR is not only the target of rapamycin and therefore the major molecular substrate that mediates the lifespan-prolonging effects of manipulation TOR activity, but also it acts as a central hub in the control of cell metabolism. In the fruit fly [7,41], similar to other models such as mice [42], pharmacological as well as genetic inhibition of TOR signaling increases lifespan substantially. Moreover, TOR signaling is insofar highly relevant as it is believed to be one of the major substrates responsible for transducing dietary restriction into lifespan prolongation [43]. Tightly interwoven with TOR signaling is the insulin-signaling pathway [44,45], which is also known to play a central role for lifespan determination [46]. Application of the BAE had no effect on the levels of circulating dILP2, which is a major insulin-like peptide in *Drosophila*. Nevertheless, an indirect effect on insulin-signaling is highly probable, because the transcript levels of *Imp-L2,* a highly potent inhibitor of insulin signaling, are increased substantially after prolonged confrontation with brown algae extracts. Reduction of effective insulin signaling can thus be achieved through two independent mechanisms: 1) the control of insulin release directly, and 2) the enhanced release of *Imp-L2*, mainly from peripheral sites. This implies that insulin signaling might be reduced indirectly via this mechanism that is independent on direct control of insulin release [18,47,48]. Our experiments not only revealed the central role of TOR signaling for transducing the positive effects of brown algae extract on lifespan, but also the central role of body fat for mediating these effects. Body fat, together with the brain, is thought to be the major organ responsible for transducing various dietary restriction-associated interventions on lifespan [19,46]. Body fat is the major energy storage organ of insects that acts itself as an endocrine organ regulating various other organs, but also insulin release from insulin producing cells in the brain [49]. Thus, reducing manipulation of TOR signaling in the fat body appears to be the mode of action that brown algae extract induces its positive effects on lifespan, implying that this extract may act as a true dietary restriction mimetic.

These types of nutritional intervention studies often suffer from confounding factors, most importantly from an unwanted induction of a dietary restriction phenotype caused either by reduced food intake [50] or by reduced uptake capacities in the intestine [51]. Rapamycin, for example, caused reduced food intake in honeybees, which may account, at least in part, for the observed lifespan extension [50]. Several cases of interferences with uptake processes in the intestine by plant-derived products have been reported [51,52,53,54]. Examples for this effect are different flavonoids that effectively inhibit glucose uptake [51], or green tea extracts that impair lipid uptake into intestinal enterocytes [53]. Another hallmark of dietary restriction is a pronounced effect on fitness that unfolds as a substantially reduced fitness, meaning reduced fecundity [55]. The reason for this reduced fitness is still not fully understood; it may follow the disposable soma theory or simply result from the reduced resources present in general, thus leading to reduced investments in reproduction [56,57]. As food intake and fecundity are not negatively affected at all in response to brown algae extract consumption, we could exclude any confounding effect caused by an unwanted dietary restriction. In contrast, low BAE concentrations increased fecundity rather than decreased it. Thus, DR or calorie restriction (CR) related effects could be excluded. Since we can exclude these simple explanations underlying the effects of BAE on lifespan extension, alternative hypotheses have to be considered. A huge number of studies claiming a life lengthening effect of certain compounds/extracts have been published, most of which were purely descriptive, giving no indication regarding the underlying mechanisms, and thus always being at risk to measure a dietary restriction-like phenotype induced by the compound/extract under investigation [51,53]. Moreover, an antioxidant activity was supposed to underlie life-prolonging effects of various extracts [58,59], but this has to be interpreted cautiously, as the addition of non-enzymatic antioxidants appears to be mostly ineffective [60,61].

To identify compounds or nutritional additives with a lifespan prolonging effect, high-throughput or semi high-throughput screening systems are mandatory [14,62]. These systems critically depend on suitable and short-lived model organisms that can be adapted to these higher throughput approaches. Until now, only yeast, *C. elegans*, and *Drosophila* have successfully and reproducibly been used for this purpose [11,13,14,63]. Although *C. elegans* is the most popular model in screening approaches aiming to identify lifespan-prolonging interventions, it is not ideally suited to study nutritional interventions comprising the addition of plant or algae extracts to the food, as they completely rely on bacteria as a food source. In contrast, the fruit fly *Drosophila* shares substantial similarities regarding organ composition, physiology, and feeding mechanisms with humans. Interestingly, the first study showing lifespan enhancing effects of nutritional components dates back almost 70 years [13,63]. With respect to choosing the most suitable *Drosophila* strain for our studies, we used the w^1118^ strain, which is the genetic background of the vast majority of all transgenes produced in the *Drosophila* field. It has been used in great number of different studies focusing on lifespan studies; these advantages are not outweighed by disadvantages, such as neurological disorders occurring in this strain [64]. Moreover, we decided to focus the study on effects induced exclusively in adults to avoid confounding effects during development.

Taken together, we show that the brown algae extract lengthened life by interfering with TOR signaling, presumably in the body fat. Thus, the BAE represents a valid candidate for a dietary restriction mimetic.

## 5. Conclusions

The aim of the study presented was to identify plant- or algae extracts that have a life-prolonging effect. For this purpose, we used the fruit fly *Drosophila melanogaster* as a model. We found that an aqueous extract of the brown algae *Saccorhiza polyschides* mediates a life prolongation of about 10%. The positive effects become even more apparent during stress treatments. We were able to exclude the analysis of interfering effects, such as those induced by a forced caloric restriction. Furthermore, the extract seems to exert its positive effects through interaction with the TOR signaling pathway. A substantial effect on the insulin signaling pathway could also be observed. In summary, we assume 435 that the brown algae extract could also be a promising dietary supplement in humans.

## Figures and Tables

**Figure 1 nutrients-12-01172-f001:**
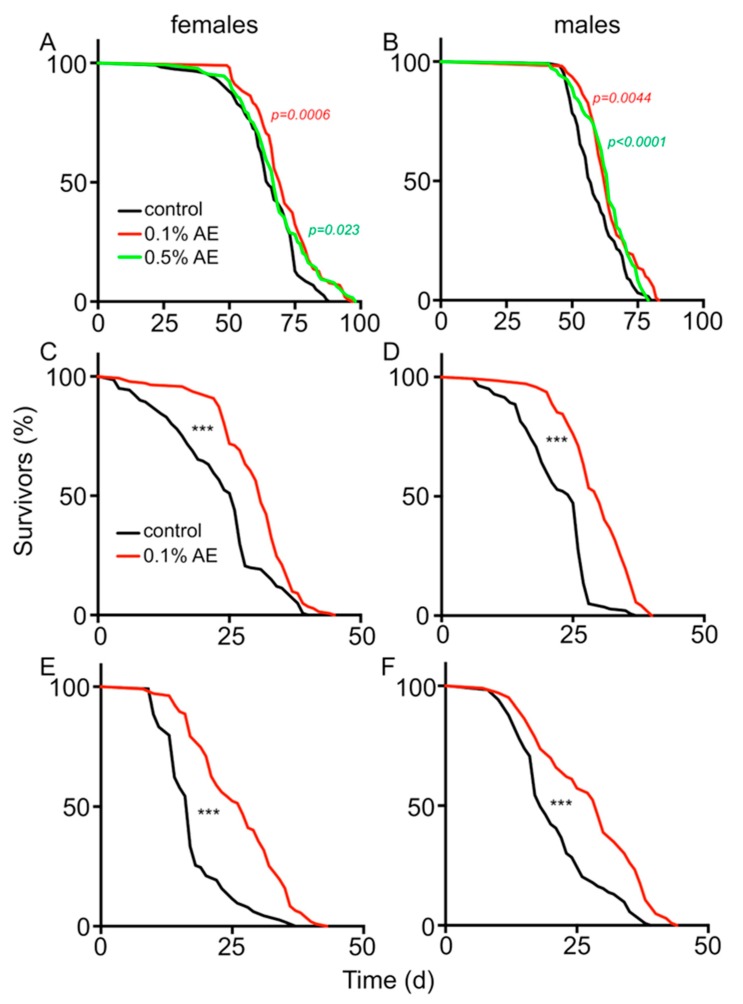
Lifespans of female (**A**) and male flies (**B**), subjected to normal food (black line), normal food supplemented with 0.1% brown algae extract (BAE) (red line), or 0.5% BAE (green line), were quantified. Lifespan analyses of female (**C**,**E**) and male flies (**D**,**F**) on control food (black lines) or on normal food supplemented with 0.1% BAE (red lines). Food was additionally supplemented with 10% triglycerides for the entire time of the experiment (**C**,**D**) or just for the first 10 days, where after flies were transferred to corresponding media without additional fat (**E**,**F**). Statistical analyses were done using a log-rank test. *n* > 100, *** means *p* < 0.001.

**Figure 2 nutrients-12-01172-f002:**
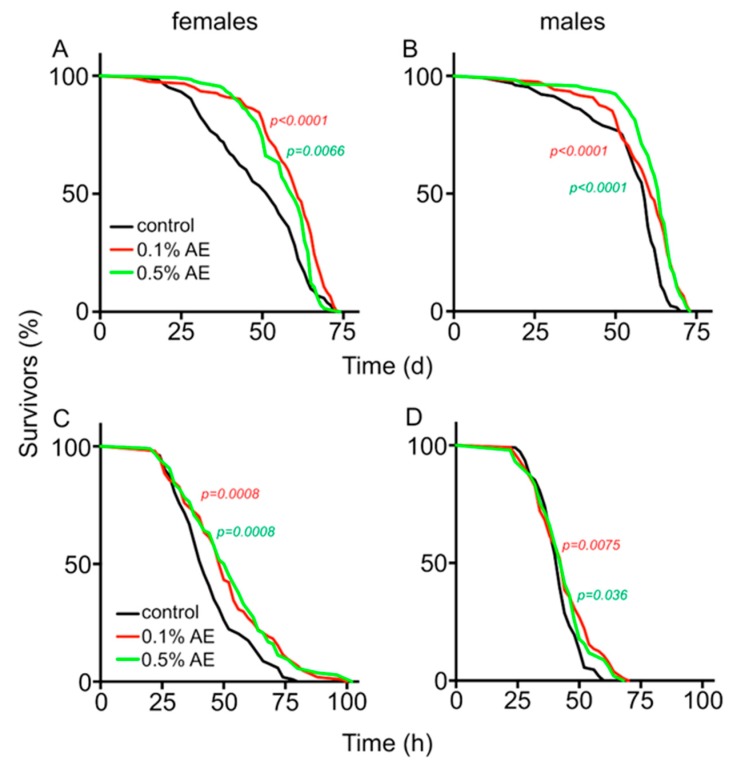
Lifespan analyses of female (**A**) and male flies (**B**), subjected to continuous drought (20% humidity) and normal food (black line), normal food supplemented with 0.1% BAE (red line), or 0.5% BAE (green line), were quantified. Statistical analysis (see Figure 1, *n* > 100 for each experiment). Starvation resistance of female (**C**) and male (**D**) flies that were fed on normal media (black lines) or on normal food supplemented with 0.1% BAE (red lines) or 0.5% BAE (green lines). Statistical analysis (see Figure 1, *n* > 100 for each experiment).

**Figure 3 nutrients-12-01172-f003:**
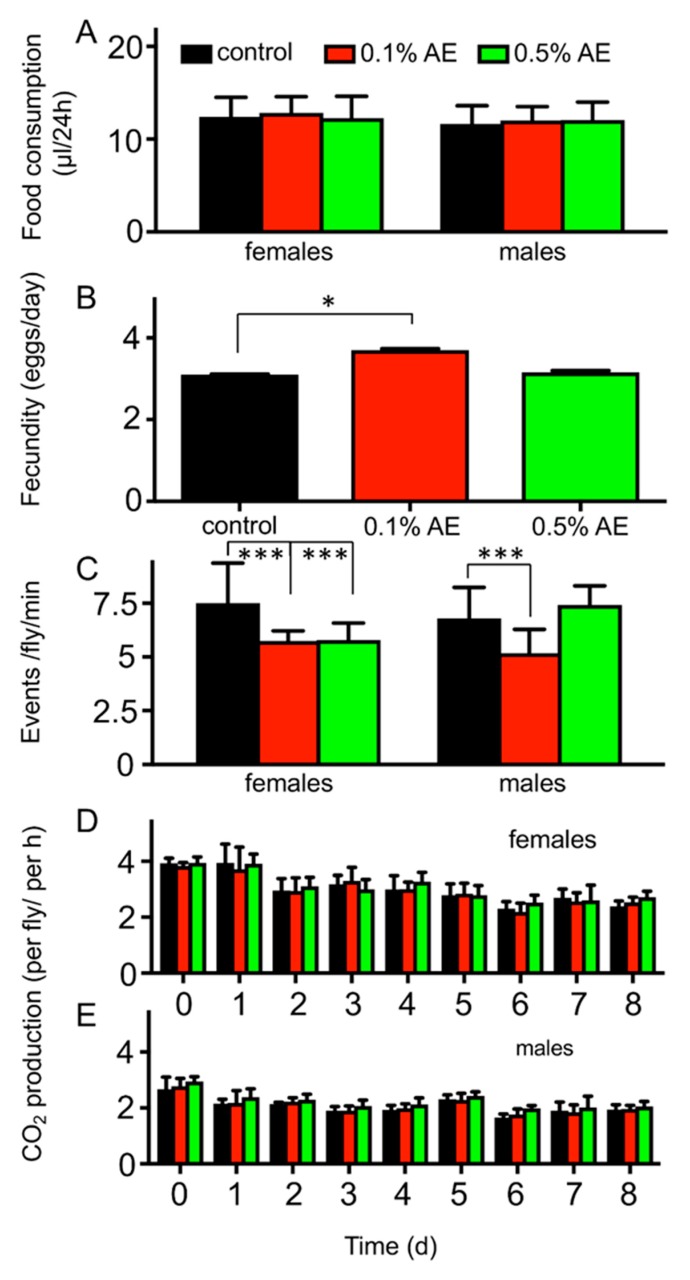
Quantification of food intake measured with a CAFE assay in female (left) and male (right) flies (**A**). Lifetime fecundity (**B**) of female flies subjected to normal food (black bars) or on normal food supplemented with 0.1% BAE (red bars) or 0.5% BAE (green bars). Overall physical activities (**C**) were quantified using the *Drosophila* Activity Monitor (DAM) system in a 24 h period for females (left) and males (right). Metabolic rates were measured in small cohorts of flies (3 per biological replicate), both in females (**D**) and males (**E**) subjected for the indicated times to the different media (control medium, black bars; 0.1% BAE, red bars; 0.5% BAE, green bars). Mean values ± SD are shown. *n* > 5 for each experiment. * *p* < 0.05, *** *p* < 0.001.

**Figure 4 nutrients-12-01172-f004:**
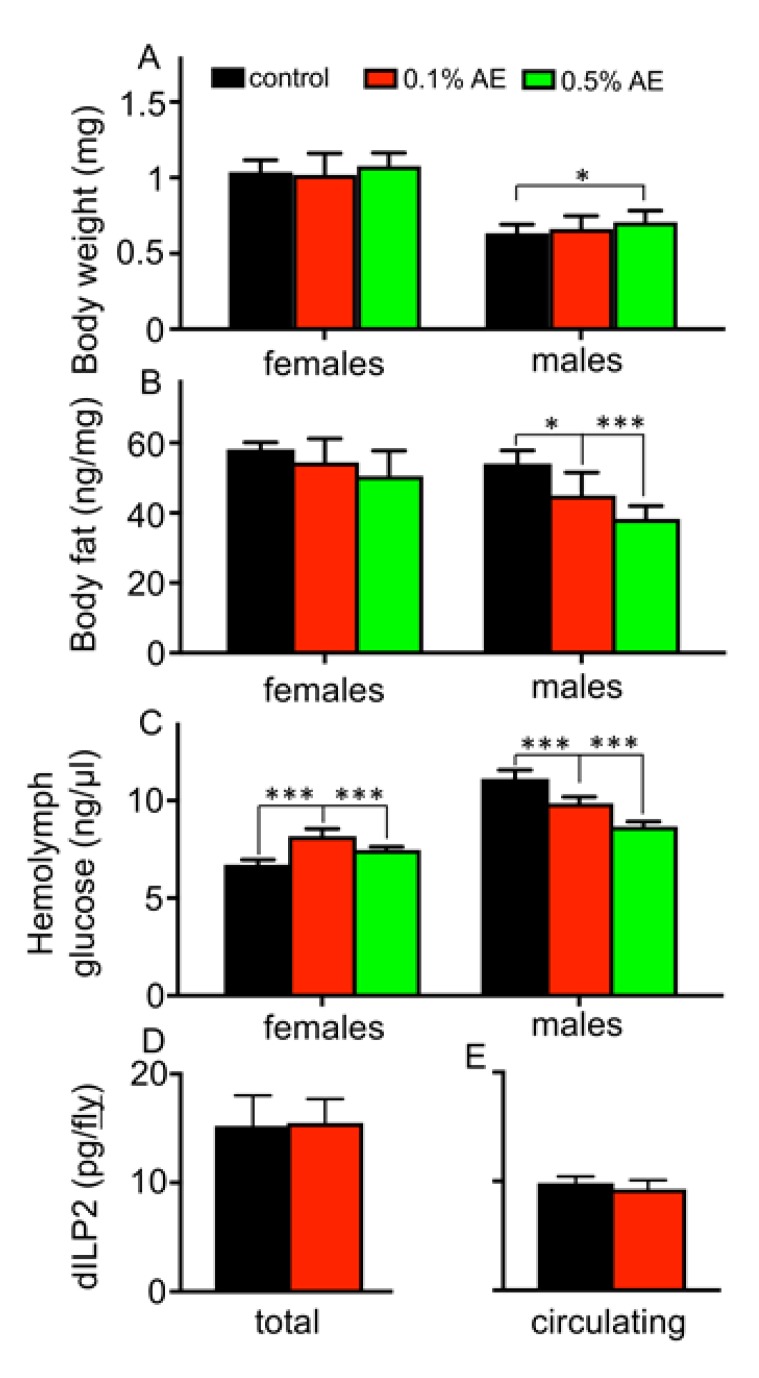
The body weights (**A**), the body fat contents (**B**), and the hemolymph glucose levels (**C**) of female (left) and male flies (right) hold on control food (black bars) or on normal food supplemented with 0.1% BAE (red bars) or on normal food supplemented with 0.5% BAE (red bars). Total dILP2 levels of female flies after one week on 0.1% BAE (**D**) and those present in the hemolymph (**E**) are shown. Mean values ± SD are displayed. *n* > 5 for each experiment. * *p* < 0.05, *** *p* < 0.001.

**Figure 5 nutrients-12-01172-f005:**
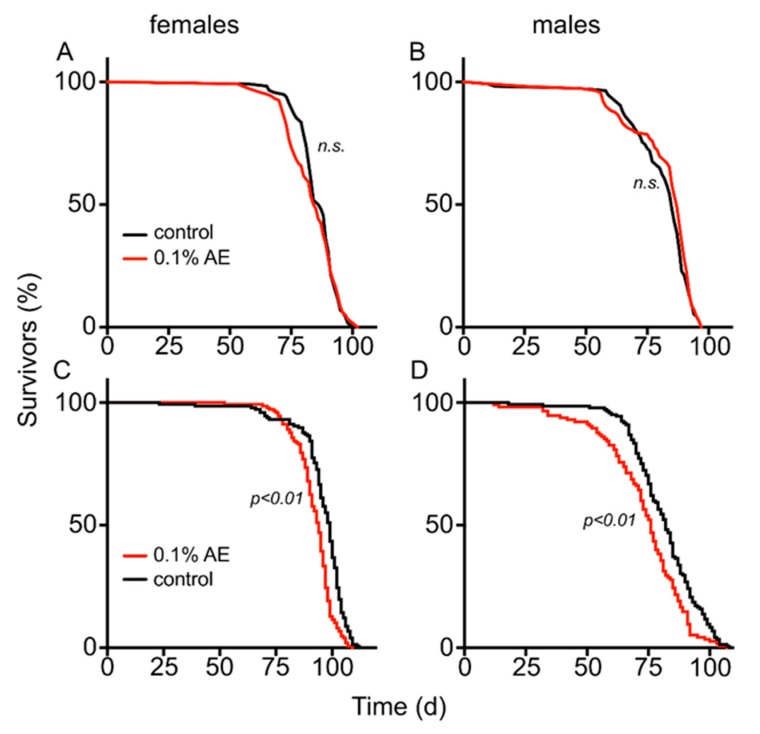
Effect of BAE on lifespan in TOR-deficient animals. TOR-deficient females (**A**) and males (**B**) were analyzed with regards to their lifespans. Control food (black lines) and those subjected to control food plus 0.1% BAE were analyzed. Effect of algae extract (AE) on lifespan in animals with overexpression of a dominant negative isoform of TOR (**C**,**D**) specifically targeted to the fat body, using the hormone-inducible GeneSwitch system in females (**C**) and males (**D**). Experimental animals induced by addition of the hormone only hold on normal food (black lines) and those subjected to 0.1% AE are shown. *n* > 100 for each experiment. *p*-values are given in the figures.

**Figure 6 nutrients-12-01172-f006:**
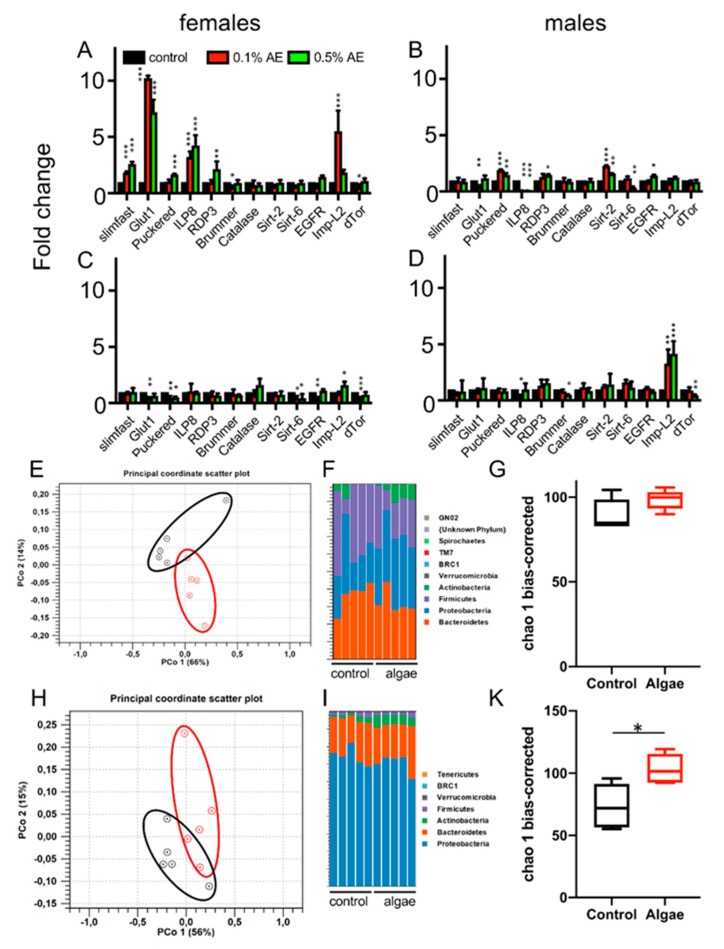
Transcript levels of selected genes in females (**A**,**C**) and males (**B**,**D**) subjected to 7 d of BAE-supplemented food (**A**,**B**) or 14 d of BAE-supplemented food (**C**,**D**); 0.1% AE (red bars) and 0.5% AE (green) are shown together with matching controls. Effects of BAE on the composition of the intestinal microbiota of w^1118^ animals (**E–K**). Analyses of 16S rRNA genes of isolated intestine (**E–G**) and of fecal samples (**H–K**) were done with 5 biological replicates for both controls and those treated with 0.1% BAE, respectively. PCoA analysis of the samples derived from the intestinal (**E**) and fecal (**H**) samples. Composition of the intestinal (**F**) and fecal (**I**) microbiota is given. The species richness (alpha diversity) is shown in (**G**) and (**K**) for intestinal and fecal samples, respectively. *n* = 5, * *p* < 0.05, ** *p* < 0.01, *** *p* < 0.001.

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
