# Peer review of "Furbellow (Brown Algae) Extract Increases Lifespan in Drosophila by Interfering with TOR-Signaling"

_nutrients, 2020, doi:10.3390/nu12041172_

Round 1
Reviewer 1 Report
Li Y et al., analyzed the effects of Furbellow (brown algae) on the lifespan of Drosophila. They found that the administration of brown algae extracts increases the lifespan of Drosophila by affecting insulin and TOR signaling.
Major points
1) The authors should describe which statistical analysis the authors used for Fig. 3, 4 and 6.
2) In Figure 5, the authors used TOR-deficient animals generated by the GeneSwitch system. However, no specific information for these experiments was described. For example, how did the authors make TOR-deficient fruit fly? How did the authors overexpress the dominant-negative isoform of TOR? How did the authors confirm the tissue-specific expression of TOR?
3) In line 229-242, the authors described that they performed rapamycin experiments, and concluded that no differences were observed. The authors should describe how they perform rapamycin experiments. Which concentration was used? In addition, the authors should show the results (not just sentences).
4) The authors should describe the approved number of this study in the Materials and Methods section.
Minor points
5) In figure legend, the authors used a small letter for figure number (a, b, c, etc...) but a capital letter was used in the Figure.
Author Response
Dear reviewer,
below you will find a point-by-point reply to your comments.
Reviewer 1
Comments and Suggestions for Authors
Li Y et al., analyzed the effects of Furbellow (brown algae) on the lifespan of Drosophila. They found that the administration of brown algae extracts increases the lifespan of Drosophila by affecting insulin and TOR signaling.
Major points
- The authors should describe which statistical analysis the authors used for Fig. 3, 4 and 6.
A description of the statistical tests used for the figures is now included in the material and methods section (lines 106-109).
2) In Figure 5, the authors used TOR-deficient animals generated by the GeneSwitch system. However, no specific information for these experiments was described. For example, how did the authors make TOR-deficient fruit fly? How did the authors overexpress the dominant-negative isoform of TOR? How did the authors confirm the tissue-specific expression of TOR?
Tor deficient flies, as well as the other genotypes, are now described in detail. The lines used have successfully been used earlier (e.g. Hoffmann et al. Aging 2013). A more detailed description of the genotypes used in this study has now been included in the Material and Methods section (lines 77-81).
3) In line 229-242, the authors described that they performed rapamycin experiments, and concluded that no differences were observed. The authors should describe how they perform rapamycin experiments. Which concentration was used? In addition, the authors should show the results (not just sentences).
We deleted the rapamycin part from the manuscript, as it gave no extra information.
4) The authors should describe the approved number of this study in the Materials and Methods section.
Numbers are now always included
Minor points
5) In figure legend, the authors used a small letter for figure number (a, b, c, etc...) but a capital letter was used in the Figure.
This has been changed throughout the manuscript.
Reviewer 2 Report
Overall this manuscript is well written and organized and the scientific question is addressed well. I have some comments below that I think some would help clarity and some are questions about the analysis that could be included to help readers better understand the project. I have used the line numbers to identify comments on specific areas of the manuscript.
Line 77: Fly Husbandry - Using W1118 is well justified. However the remaining Drosophila genetics are missing including the genetics of the UAS/Gal4 lines used, the TOR transgenes, and the dILP tagged fly lines. Which stocks were utilized, citations of characterization and validation of these lines, and where these lines are available (if applicable) should be added to this section. This would also help to better understanding the development timing of TOR disruption.
Line 81: The methodology of life span is well written. It is unclear why the authors chose just to have dietary changes on adults and not use larvae. An explanation of this would be helpful.
Line 110: Was the metabolic rate determination controlled for time of day? It would seem possible that metabolic rate may vary depending on time of day and circadian rhythm.
Line 123: See comments on fly husbandry to better define the genetics of the lines used for dILP quantification.
Line 160: Figure 1 - is there any data on what would happen if flies started on a normal diet and then shifted to the BAE diet after a period of time? This data may be more relevant to humans, as humans tend to shift dietary habits as they get older.
Line 209: Authors state that BAE has no impact on fecundity, but figure B shows an increase of fecundity from control to 0.1%. This difference should be addressed and discussed. Additionally, it is unclear if this measure of fecundity is based on unfertilized eggs or fertilized eggs based on the methodolgy.
Line 234: There is a mention of rapamycin treated flies and BAE, is there a reason this data does not seem to shown? Also why was this only done with 0.5% BAE and not also 0.1% as the other experiments.
Line 237: Lifespan analysis was performed at elevated temperatures. What temperatures and what was the reason for this change? If it was to further drive the changes in TOR signaling due to the UAS/Gal4 temperature sensitivity than that should be stated.
Line 244: Figure 5. This figure need statistics to mirror that in the other figures. It seems that conducting statistical analysis on this type of experiment is feasible and should be consistent throughout the manuscript.
Line 252: How were these target genes selected for study? Were there any changes to overall RNA content in the animals?
Line 278: The second sentence of the discussion is a bit hard to follow (at least for me) and could be written to better clarify the distinction between nutrient and calorie intake.
Line 299: The authors used a dominant negative TOR and demonstrated changes in lifespan. What would be the expectation in an over-expression TOR system?
Line 310: There are shown changes imp-L2 but no changes in circulating dILP2. This seems contradictory and this should be addressed in the discussion.
Line 344: There is a line break between paragraphs here in the discussion that should be removed.
Author Response
Dear reviewer,
below you will find a point-by-point reply to your comments.
Overall this manuscript is well written and organized and the scientific question is addressed well. I have some comments below that I think some would help clarity and some are questions about the analysis that could be included to help readers better understand the project. I have used the line numbers to identify comments on specific areas of the manuscript.
Line 77: Fly Husbandry - Using W1118 is well justified. However the remaining Drosophila genetics are missing including the genetics of the UAS/Gal4 lines used, the TOR transgenes, and the dILP tagged fly lines. Which stocks were utilized, citations of characterization and validation of these lines, and where these lines are available (if applicable) should be added to this section. This would also help to better understanding the development timing of TOR disruption.
This has now been included in the materials and methods section (lines 77-81). A detailed description is now added in the material and Methods section (lines 130-140).
Line 81: The methodology of life span is well written. It is unclear why the authors chose just to have dietary changes on adults and not use larvae. An explanation of this would be helpful.
We have included this (lines 452-453). Usually, dietary interventions are restricted to adults to avoid any confounding effects (either positive or negative) caused by the algae extracts on the development of the flies.
Line 110: Was the metabolic rate determination controlled for time of day? It would seem possible that metabolic rate may vary depending on time of day and circadian rhythm.
Yes, it was controlled for time of the day to exclude confounding effects caused by potentially different metabolic rates controlled by circadian activities.
Line 123: See comments on fly husbandry to better define the genetics of the lines used for dILP quantification.
This has now been included (lines 77-81).
Line 160: Figure 1 - is there any data on what would happen if flies started on a normal diet and then shifted to the BAE diet after a period of time? This data may be more relevant to humans, as humans tend to shift dietary habits as they get older.
You are completely right, but we decided to focus on constant confrontation, which is also the experimental strategy used in most nutritional interventions (e.g. dietary or caloric restriction) (lines 84-85).
Line 209: Authors state that BAE has no impact on fecundity, but figure B shows an increase of fecundity from control to 0.1%. This difference should be addressed and discussed. Additionally, it is unclear if this measure of fecundity is based on unfertilized eggs or fertilized eggs based on the methodolgy.
We now include a short discussion of this result (lines 424-427). Counted were the eggs, but no difference in the hatching rates were observed.
Line 234: There is a mention of rapamycin treated flies and BAE, is there a reason this data does not seem to shown? Also why was this only done with 0.5% BAE and not also 0.1% as the other experiments.
We removed this small part of the manuscript because no additional information is given.
Line 237: Lifespan analysis was performed at elevated temperatures. What temperatures and what was the reason for this change? If it was to further drive the changes in TOR signaling due to the UAS/Gal4 temperature sensitivity than that should be stated.
We removed the sentence dealing with the rapamycin treatment because it gave no extra information.
Line 244: Figure 5. This figure need statistics to mirror that in the other figures. It seems that conducting statistical analysis on this type of experiment is feasible and should be consistent throughout the manuscript.
Statistics are included in the text and they are now also part of the figure.
Line 252: How were these target genes selected for study? Were there any changes to overall RNA content in the animals?
We choose especially those genes that are involved in either Tor-, insulin- or JNK signaling. In addition, we choose genes that are relevant for uptake and storage of nutrients, and some (sirtuins) that have been shown to be involved in the response to CR. We did not analyze the total amount of RNA that could be extracted from the flies, but noted no big differences, implying that RNA levels remained comparable. We have to admit that no data are available.
Line 278: The second sentence of the discussion is a bit hard to follow (at least for me) and could be written to better clarify the distinction between nutrient and calorie intake.
You are absolutely right, it is indeed hard to understand and we have changed this accordingly.
Line 299: The authors used a dominant negative TOR and demonstrated changes in lifespan. What would be the expectation in an over-expression TOR system?
Overexpression of the TOR system should lead to (and it indeed leads) an increased lifespan. Under these experimental conditions, BAE should not lead to increased lifespan as the TOR pathway is activated to a high degree.
Line 310: There are shown changes imp-L2 but no changes in circulating dILP2. This seems contradictory and this should be addressed in the discussion.
We now added this to the discussion section – we are aware of this seemingly contradictory result, but this might reflect two independent ways to reduce effective insulin signaling, one via release control and one peripherally controlled way, through the release of ImpL2.
Line 344: There is a line break between paragraphs here in the discussion that should be removed.
This line break has been removed.
Reviewer 3 Report
it is a interesting and well-conducted study.
Author Response
Dear reviewer.
thanks a lot
Round 2
Reviewer 1 Report
- The authors should describe which statistical analysis the authors used for Fig. 3, 4 and 6.
A description of the statistical tests used for the figures is now included in the material and methods section (lines 106-109).
The authors added the description of the statistical tests. Because Student's t-test and Mann-Whitney test were not suitable for multiple comparisons, the authors should use Dunnet, Bonferroni or Tukey test.
4) The authors should describe the approved number of this study in the Materials and Methods section.
Numbers are now always included
I could not find any description of the approved number of this study. Because the author used genetically modified fruit flies, this study may be approved by an ethical review committee.
Author Response
Dear reviewer,
sorry for the misunderstanding. We have performed Dunnett' test for the corresponding figure (3, 4, 6) and included modified versions of the figures. Moreover, we added this information to the Material and Methods section and slightly corrected the results section accordingly.
Again sorry for the misunderstanding, but for experiments with Drosophila, even with transgenic ones, no ethical approval is required.
Best regards
Thomas Roeder